# Creating Acceptable Tablets 3D (CAT 3D): A Feasibility Study to Evaluate the Acceptability of 3D Printed Tablets in Children and Young People

**DOI:** 10.3390/pharmaceutics14030516

**Published:** 2022-02-25

**Authors:** Louise Bracken, Rober Habashy, Emma McDonough, Fiona Wilson, Joanne Shakeshaft, Udeme Ohia, Tamar Garcia-Sorribes, Abdullah Isreb, Mohamed A. Alhnan, Matthew Peak

**Affiliations:** 1Paediatric Medicines Research Unit, Alder Hey Children’s NHS Foundation Trust, Liverpool L12 2AP, UK; louise.bracken@alderhey.nhs.uk (L.B.); emma.mcdonough@alderhey.nhs.uk (E.M.); joanne.shakeshaft@alderhey.nhs.uk (J.S.); 2School of Medicine and Biomedical Science, University of Central Lancashire, Preston PR1 2HE, UK; rhabashy@uclan.ac.uk (R.H.); tgracia-sorribes@kcl.ac.uk (T.G.-S.); aisreb@uclan.ac.uk (A.I.); 3NIHR Alder Hey Clinical Research Facility, Alder Hey Children’s NHS Foundation Trust, Liverpool L12 2AP, UK; fiona.wilson@alderhey.nhs.uk (F.W.); udeme.ohia@kcl.ac.uk (U.O.); 4Centre for Pharmaceutical Medicine Research, Institute of Pharmaceutical Science, King’s College, London SE1 9NH, UK

**Keywords:** additive manufacturing, personalised medicine, patient-specific, age-appropriate, point-of-care manufacturing

## Abstract

3D printing (3DP) has been proposed as a novel approach for personalising dosage forms for children and young people (CYP). Owing to its low cost and the lack of need for finishing steps, fused deposing modelling (FDM) 3DP has been heavily researched in solid dosage forms (SDFs) manufacturing. However, the swallowability and overall acceptability of 3D printed dosage forms are yet to be established. This work is the first to evaluate the acceptability of different sized 3D printed placebo SDFs in CYP (aged 4–12 years). All participants had previously participated in a feasibility study (CAT study) that assessed the swallowability and acceptability of different sized GMP manufactured placebo conventional film-coated tablets, and therefore only attempted to swallow one 3D printed tablet. The participants assessed the swallowability, acceptability, mouthfeel, volume of water consumed, and taste of the sample using a 5-point hedonic facial scale on a participant questionnaire. A total of 30 participants were recruited, 87% of whom successfully swallowed the 3D printed tablet that they attempted to take. Attributes of the 3D printed tablets were scored as acceptable by the following percentage of participants—swallowability (80%), mouthfeel/texture (87%), the volume of water consumed (80%), taste (93%), and overall acceptability (83%). Overall, 77% of children reported they would be happy to take the tablet every day if it was a medicine. Participants were also asked which tablets felt better in the mouth—the film-coated tablets or the 3D printed tablets, and the most popular response (43%) was that both were acceptable. This study shows that FDM-based 3D printed SDFs may be a suitable dosage form for children aged 4–12 years. The results from this feasibility study will be used to inform a larger, definitive study looking at the acceptability of 3D printed tablets in children.

## 1. Introduction

The administration of medicines to children and young people (CYP) poses a challenge to many parents and healthcare professionals. Medication adherence rates in CYP range from 11% to 93% [1]. Developing medicines that are acceptable to CYP have the potential to influence adherence to therapeutic regimens and improve treatment outcomes [2]. Acceptability has previously been defined as “an overall ability of the patient and caregiver (defined as “user”) to use a medicinal product as intended” [3]. Historically, oral liquids were generally regarded to be the most appropriate dosage form for children. However, challenges such as taste masking, portability, stability, and the inclusion of excipients are all factors to be considered in the selection of an appropriate dosage form for children [4,5]. For children, personalising dosage forms is complicated by the need to optimise the dose, ability to swallow, mouthfeel, and taste of the product. The need for variable doses is often managed by a common practice of dose modification by healthcare professionals, parents/carers, and CYP [6]. For SDFs in particular, tablet-splitting is a common form of dose modification performed in the healthcare and home settings to achieve an intended dose for CYP [6,7,8]. Such an approach often results in dose imprecision and dose variability [9]. To achieve dose flexibility, several technologies have been developed such as microparticles, in situ gels, minitablets, pellets, and orodispersible films [10]. Klingmann et al. compared the acceptability of coated and uncoated 2 mm mini-tablets to syrup in children aged 6 months to 5 years [11]. Uncoated mini-tablets had significantly higher acceptability when compared to syrup, and were deemed a suitable alternative to liquid formulations. Another study investigated the acceptability of 3-mm mini-tablets in children aged 2–6 years [12]. Interestingly, 46% of children (<3 years old) were able to swallow the dosage form, and 85% in children (≥3 years old).

Tablets have been proposed as a suitable alternative to liquids for children, and they can overcome some of the challenges described above. The evidence for the swallowability and overall acceptability of tablets in CYP is growing, and additional studies are required [13,14]. A challenge with a standard tablet is that it cannot be titrated at the point of use, and thus manipulation at the point of administration to attempt to obtain the required dose can often lead to over or underdosing, affecting the drug efficacy and causing adverse drug reactions.

While several pre-clinical assessments for using 3D printing (3DP) to manufacture SDFs intended for CYP have been reported, there are no reports of any direct administration of an ingestible 3D printed tablet to CYP. For instance, fused deposition modelling (FDM) 3DP has been used to manufacture minitablets to deliver an accurate dose of a spasmolytic drug, baclofen with immediate release properties [15]. The taste-masking performance of indometacin tablets produced via FDM 3DP in a paediatric formulation was also assessed [16]. Novel dosage form concepts were enabled using 3DP technologies. For instance, Rycerz et al. employed embedded 3DP to fabricate a Lego^TM^ chewable dosage form for simultaneous delivery of paracetamol and ibuprofen [17]. Using a 3D printer for chocolate, hydrophilic and lipophilic active compounds were incorporated in chocolate-based dosage forms [18]. One unique example of a paediatric application was using direct extrusion for producing chewable flavoured isoleucine in children with maple syrup urine disease [19]. The innovative approach provided equivalent but less variable control of isoleucine levels in the plasma with mean levels closer to the target value.

Limited information is available on the acceptability of 3D printed dosage forms in CYP. Januskaite et al. investigated the visual preference in children based on appearance, perceived taste, texture, and familiarity [20]. While children showed a visual preference to tablets produced using digital light 3DP, their preference shifted to semi-solid extrusion-based tablets when they were informed that these are chewable. However, the swallowability of the tablets was not assessed. In another example, indometacin tablets were produced via FDM 3DP in a paediatric formulation, and their taste-masking performance was assessed in adults (18–55 years) [16]. As far as the authors know, there have been no swallowability reports of FDM 3D printed tablets in CYP.

We have previously assessed the key parameters needed to design a definitive trial of the acceptability of different-sized placebo tablets in children [21]. In this study, we aimed to establish for the first time if it is possible to administer an ingestible 3D printed tablet to CYP aged 4–12 years. In addition, we estimated a range of variables of the acceptability to CYP of 3D printed tablets. We also aimed to identify if a difference in acceptability is found between standard round biconvex and equivalent 3D printed tablets.

## 2. Materials and Methods

### 2.1. Materials

The excipients used include Eudragit EPO provided from Evonik Industries (Darmstadt, Germany). Sodium phosphate fumarate (PRUV) was donated from JRS Pharma, and titanium dioxide was obtained from Special Ingredients (Chesterfield, UK). Talc (Luzenac Pharma M) was donated by Imerys Talc (Paris, France). Triethyl Citrate (TEC) FCC was purchased from Sigma-Aldrich (Poole, UK).

### 2.2. Filament Production

All the preparation for filament and 3D printed tablets was carried out in a dedicated clean area. The hot-melt extrusion (HME) machine was cleaned using a specially designed protocol. Approximately 10 g of materials (Eudragit EPO:triethyl citrate:talc:TiO_2_ weight ratio of 45:5:49:1) were added gradually to a counter flow twin-screw hot melt extruder equipped with a 1.75 mm nozzle, HAAKE MiniCTW (Karlsruhe, Germany). To allow a homogeneous distribution of the powders, the molten mass was mixed in the extruder. The specific temperature of initial feeding and extrusion for the filament were 100 and 90 °C, respectively. A torque control of 0.8 Nm was used to extrude the filaments. All filaments were stored in sealed plastic bags at room temperature before FDM 3DP.

### 2.3. Design and 3DP of Tablets

Tablets were constructed with the pre-prepared filaments using a commercial FDM 3D printer equipped with a 0.4 mm nozzle size. The templates used to print the tablets were designed in caplet shape using Autodesk^®^ 3ds Max^®^ Design 2012 software version 14.0 (Autodesk, Inc., San Rafael, CA, USA). The design was saved in a stereolithography (stl) file format and was imported to the 3D printer software, MakerWare Version 2.4.0.17 (Makerbot Industries, LLC, New York, NY, USA). The 3D printed placebo tablet diameters chosen for this feasibility study were 6, 8, and 10 mm in a standard round bi-convex shape. The 3D printer was modified by replacing a standard copper nozzle with a specially designed 0.4 mm stainless steel nozzle. The approximate weights of each tablet were 86, 203, and 339 mg for tablets of 6-, 8- and 10-mm diameter, respectively.

For comparison, tablets of the same size and shape were produced to current good manufacturing practice (cGMP) standards by Quotient Sciences. Tablets were manufactured via direct compression and based on microcrystalline cellulose and coated using flavour-free Opadry Clear coating (Colorcon, Dartford, UK).

### 2.4. Heavy Metal Assay

The heavy metal contents of the 3D printed tablets were assessed using inductively coupled plasma mass spectrometry (ICP-MS) Thermo Fisher X series I equipped with an autosampler (Thermo Fisher Scientific Inc., Horsham, UK), and results were analysed using Qtegra^TM^ Version 2.7 (Thermo Fisher Scientific Inc., Horsham, UK). The auxiliary and nebulizer gas flow were both 0.5 L/min. The radio frequency (RF) power was set at 1150 W.

Samples of raw materials, filaments, and 3D printed tablets (equivalent to 1 g) were dissolved in 50 mL 1% nitric acid and 100 mL Type I water in a 250 mL volumetric flask. The components were heated to a boiling point, and the volume was completed to 250 mL after cooling using Type I water. A calibration curve was prepared at concentrations: 20, 50, 100, 200, 500, 1000, and 2000 ppm of Pb, Hg, As, V, Cu, Co, Cd, and Ni ICP/AAS standards. The outcome of each sample was calculated using the calibration curve. Each sample was assayed in triplicate.

### 2.5. Residual Solvents Studies

To assess the solvent residues of acetone, samples (100 mg tablet) were dissolved in 1000 Type I water and injected in an overhead Thermo Fisher Focus gas chromatography (GC) system. Results were analysed using xCalibur software version 2.2 (Thermo Fisher Scientific Inc., UK). The injection volume was 1 µL, the inlet and detector temperatures were 200 and 250 °C, respectively. The flow rate was 1 mL/min using helium as a carrier gas and Stabilwax GC column (Restek, Bellefonte, PA, USA).

### 2.6. Acceptability Studies

A feasibility study was performed to investigate the acceptability of different-sized placebo 3D printed tablets following the same methodology as the CAT study [21]. The acceptability was assessed in terms of swallowability, taste, mouthfeel, and the volume of water consumed to swallow each 3D printed tablet.

#### 2.6.1. Setting

The study took place in the National Institute for Health Research (NIHR) Alder Hey Clinical Research Facility (CRF) for Experimental Medicine and inpatient wards within a Regional Paediatric Hospital and a District General Hospital (DGH), in addition to outpatient clinics within the Regional Paediatric Hospital.

#### 2.6.2. Participants

The participants were CYP aged 4–12 years, able to read and/or understand English with no known difficulty swallowing and no allergy to peanuts, gluten and/or lactose. Both patients (children under the care of a Regional Paediatric Hospital or DGH) and healthy volunteers were included in the study. A healthy volunteer was defined as a participant who was not an inpatient or an outpatient attending a hospital appointment on the day of participation.

#### 2.6.3. Consent

Informed consent was obtained from the parent/legal guardian, and written assent was obtained for children aged six and over who wished to participate.

#### 2.6.4. Patient and Public Involvement

CYP contributed to the design and delivery of the study. The study design was presented to the Liverpool Young Person’s Advisory Group YPAG Generation R (https://generationr.org.uk/, accessed on 21 January 2022), and specific methods were adapted in response to feedback.

#### 2.6.5. Interventions

3D printed placebo tablets (6-, 8- and 10-mm diameter with a standard round bi-convex shape) were used. Participants were required to swallow up to three 3D printed tablets, and this was dependent upon whether they had participated in the CAT study (REC Ref.: 17/NW/0410) before participating in the CAT 3D study (Figure 1). As all the children who took part had previously taken part in the CAT study, they were only required to swallow one 3DP tablet. If they successfully swallowed all three CAT study tablets, participants were given a 3D printed tablet size at random to assess. If the participant took part in the CAT study but was not able to swallow all three tablets, they were given a 3D printed tablet which was equivalent in size to the largest tablet that they successfully swallowed in the CAT study. Participants received 150 mL of water but had free access to additional water if required. Tablets were presented to the children in medicine pots and the children then self-administered the medication.

#### 2.6.6. Staff Training

All research staff members were trained by LB in methods of acceptability assessment. This ensured standardisation of the results captured as part of the researchers’ observations.

#### 2.6.7. Outcomes

Assessment of Domains of Acceptability

The study followed the same methodology for acceptability assessment as the CAT study (Bracken et al., 2020). Using the Researcher Observation Sheet, a researcher observed the participant attempting to swallow the 3D printed tablets and used a flashlight to check for any remaining tablet residue due to chewing. To minimize any potential operator bias, the clinical researchers who administered the 3D tablets were trained in how to undertake field observations of acceptability and other assessments using a standardised assessment protocol and Researcher Observation Sheet. In addition, the participants were required to complete the questionnaire independently. The clinical researchers undertaking observations were not involved in any aspect of tablet design and manufacturing.

Participants completed a short paper questionnaire to obtain their views on the swallowability of the tablet immediately after swallowing or attempting to swallow the 3D printed tablet. Acceptability was evaluated by participants using a five-point hedonic scales (Appendix A). Values of 1–3 on the hedonic scale were deemed as acceptable.

The volume of water consumed was recorded by the researcher. Participants were also asked to rate the volume of water required with each tablet administration using a hedonic scale (Appendix A) when asked “what did you think of the amount of water you had to drink?”. Participants were asked to rate the taste of the tablet swallowed using the hedonic scale as part of the questionnaire. The anchors used were “very good”, “very bad” on the hedonic scale, and numerical values of 1–3 were deemed acceptable.

Assessment of Overall Acceptability

Overall acceptability was measured by participants rating on how acceptable the tablet was to take directly following administration using a hedonic scale (Appendix A). Anchors used were “very good” to “very bad” and a score of 1–3 was deemed acceptable. Facial expressions and behaviours before, during, and after administration of the 3DP tablet were observed and recorded by a researcher using a tick box chart.

A total score was calculated based on any facial expressions and/or behaviours observed by the researcher at the time before, during, or following the 3DP tablet administration attempt (Bracken et al., 2020). Any additional verbal comments from the participant in terms of acceptability were also noted.

Finally, participants were asked: if they would be willing to take the 3D printed tablets every day; if it were medicine and to select the most important factor if they had to take medicines every day, from a list of options: taste, smell, tablet size, the taste left in the mouth or texture.

### 2.7. Regulatory Approvals

Ethical approval for this study was obtained from an NHS Research Ethics Committee (REC), REC Reference 18/NW/0019, and the Health Research Authority, IRAS No. 233808. The Medicines and Healthcare products Regulatory Agency (MHRA) confirmed it was not required to issue an MHRA Clinical Trials Authorisation as the study only involved placebo 3D printed tablets.

## 3. Results

A computer-aided design (CAD) was specially devised to mimic the shape and structure of biconvex film-coated tablets (Figure 2). As producing bi-convex tablets from both sides is difficult to achieve in FDM 3DP without additional steps, the design was modified to include a flat base. Such an approach allows the production of 3D printed tablets without the need to include a draft. Initially, tablets with different resolutions were made (data not shown), and tablets with 80 µm-layer thickness provided the smoothest surface tablet. In FDM 3D printing, a layer thickness of 200 µm is often considered as an optimal value between resolution and 3D printing time. While several FDM 3D printers have been optimised to achieve a layer thickness as thin as 30 µm, increasing resolution is associated with longer printing time-per-tablets and lowers process efficiency.

SEM images indicated the formation of smooth coating with film-coated tablets (Figure 3A). 3D printed tablets showed circular layers, and the tip on the top is where the 3D printed tablet is finished before moving to the next unit (Figure 3B). The raw materials and produced 3D printed tablets passed the ICH specifications for heavy metal contents and solvents’ residues (Appendix A). The tablets showed no or limited traces of heavy metals.

A total of 30 participants (aged 4–12 years) were recruited to swallow 3D printed tablets, following swallowing one or more film-coated placebo tablets from the CAT study (Figure 4, Table 1). Most participants were aged between 8–10 years (median age = 8.5 years, average age = 8.6 years). The 3D printed tablet was assessed and rated for mouthfeel, acceptability, the volume of water consumed, and taste. The overall acceptability score of the participants is shown in Figure 5. The attributes of the 3D printed tablets were scored as acceptable by the following percentage of participants: swallowability (80%), mouthfeel/texture (87%), the volume of water consumed (80%), acceptability (83%), and taste (93%). In general, a tablet size of size 6 mm showed the highest overall acceptability score followed by tablets sized at 8 and 10 mm.


**
*Successful swallowing and mouthfeel*
**


The ability to swallow tablets could be directly linked to the age of the volunteer and smaller tablets. All participants were able to successfully swallow 6-mm 3D printed tablets (*n* = 8). For 8-mm 3D printed tablets, nine of the 10 participants who swallowed the film-coated tablets also managed to swallow the 3D printed tablets. However, for 10 mm tablets, 75% of participants were able to swallow the 3D printed tablets. However, four of 12 participants (6-, 8-, 8- and 11-year-old) could not swallow the tablet. These participants reported that the texture of the 3D printed tablets was ‘hard’ or ’rough’ and were unable to swallow it. One participant commented that it was easier to swallow the film-coated tablets as they were smoother. Another reported that the 3D printed tablet ‘looked cool but the texture was bad’. Overall, the data suggest that 87% of participants (26 out of the 30 participants) who can swallow coated tablets can swallow tablets produced by FDM 3DP. Some children were more positive ‘It was fun, and I would do it again’, ‘It was very fun’ ‘It was different but ok’. While the size of tablets does not allow a decisive conclusion, the swallowability decreased with the larger-sized tablets (8 and 10 mm).


**
*Volume of water (n = 26)*
**


The average volume of consumed water was lower in participants aged 9–12 years: for children in the 4- to 8-year-old group, the average volume of water needed was 38.1 mL (27.4, 15.3, and 71.5 mL for 6-, 8-, and 10-mm tablets, respectively), whereas children aged 9–12 years of age consumed an average volume of 19.9 mL (9, 16.7 and 34 mL for 6, 8- and 10-mm tablets, respectively). This was in agreement with our previous findings for film-coated tablets [21], where younger children on average needed 18.2 mL more water.


**
*The overall acceptability (n = 30)*
**


The overall acceptability appears to be dependent on the size of the tablet. The overall acceptability for the 3D printed tablets across all ages of CYP studies was reported to be 100%, 90%, 67% for 6-, 8- and 10-mm tablets, respectively. The 8-mm tablet seems to be the most acceptable for 3D printed tablets; this is in agreement with our previous findings with film-coated tablets [21].


**
*Willingness to take the tablet every day (n = 30)*
**


Overall, 77% of participants reported they would be willing to take the 3D printed tablets every day if they were medicine (Figure 6 and Figure 7). This response was 100%, 80% and 58% for 6-, 8- and 10-mm tablets, respectively. The participants were also asked which tablets felt better in the mouth, the film-coated tablets or the 3D printed tablets and the most popular response (43%) was that ‘both felt ok’. However, for the 4–8-year-old participants, film-coated tablets were preferred.

## 4. Discussion

To our knowledge, this is the first study to report the administration of ingestible 3D printed tablets to CYP. In doing so, this study moved into new territory and explored the regulatory framework for conducting studies using 3DP non-GMP tablets in children. In addition, 86.7% of participants successfully swallowed the 3DP tablet they attempted to take compared to 91.5% for CAT study tablets. Attributes of the 3D printed tablets were scored as acceptable by the following percentage of participants swallowability (80%), mouthfeel/texture (87%), the volume of water required (80%), acceptability (83%), and taste (93%). Furthermore, 77% of children advised they would be happy to take the tablet every day if it were a medicine.

The data demonstrate that 3D printed SDFs, particularly 6-mm, are an acceptable option for CYP aged 4–12 years. Although the size of the sample is limited, the study also provides a comparison of the potential overall acceptability of tablets fabricated by this emerging technology in comparison to conventionally manufactured bi-convex tablets.

The size of the tablets used in this study conforms to those of conventionally sized tablets, rather than minitablets. The literature has shown that training can help children to swallow tablets of larger size than minitablets [22,23]. All the participants in this study had already demonstrated an ability to swallow a film-coated tablet in the size and shape tested in this study. Only 4 out of the 30 participants could not swallow the larger size tablets (8 and 10 mm). The difficulty of swallowing 3D printed tablets could potentially stem from the presence of layers in the tablet structure. This could result in the formation of a rougher tablet surface. Another possible contribution to the difference of the 3D printed tablets compared to the film-coated ones stems from the composition of the tablet.

In this study, there were no reports that the taste of 3DP tablets was a barrier to acceptability. In this instance, methacrylate polymer has been used. It is widely used for film coating for taste masking. Eudragit E, a cationic co-polymer based on diethylaminomethyl methacrylate and neutral methacrylic esters, has been designed as a coating taste-masking solution. The polymer is insoluble at pH > 5 to prevent undesirable drug release in the oral cavity whilst instantly dissolving in the gastric environment of the stomach. Eudragit E recently received significant interest in paediatric formulations for its taste-masking properties by coating minitablets [24,25], tablet extrusion [26], or wet granulation [27]. Being the bulk of the polymer in the matrix, it might have contributed to the positive taste score of 3D printed tablets. By using this polymer, it will be possible to incorporate bitter drugs in a formulation designed for children.

The size of the tablet appeared to have played an important role. One additional factor is the flat surface of the tablet of 3D printed tablet compared to a film-coated tablet, which allows the formation of complete bi-convex tablets. The use of FDM 3DP dictates the need for a flat surface for a base to avoid the use of draft and mitigate the risk of object movement during 3DP.

Examples of swallowing 3D printed SDFs are particularly rare. One example was reported in adults where the swallowability of polylactic acid-based FDM 3D printed tablets of different sizes and shape was assessed [28]. The report indicated that tablets of >6 mm thickness are more difficult to swallow compared to round low thickness tablets. However, the finished surface of these tablets (using sanding technique) and the age of the participants (adults) did not allow a direct comparison with our findings.

In this work, Eudragit E and talc-based matrix produced a tablet with a grey tinge. Therefore, titanium dioxide was added to give tablets a white colour to mimic tablets coated by conventional tablets coated with a commonly used Opadry^TM^ coating. Using bright white colour improved the overall cosmetic appearance of the 3D printed tablets. This could be partially attributed to making the seamlines between 3D printed layers less notable. The impact of different colours on tablet preference was not tested. Using certain colours in tablet coating may impact patient perception. For example, white-coloured tablets were favoured by patients [29]. In addition, they made the perception of a bitter flavour. In another example, a yellow colour appeared to provide the perception of a lemon/citrus flavour [20].

The main application of 3DP technology conceivably appears in the context of the need for flexible dosing [30,31,32]. While personalised medicine aims to deliver the right drug to the right patient, achieving the right dose at the right time could often be undermined. In fact, achieving personalised dose accurately and consistently through commercially available DFs has been a major challenge. In this context, 3DP offers a powerful means to timely deliver an accurate dose and mitigate the risk of manipulation of age-inappropriate formulations [33,34].

3DP has a great potential to manufacture personalised tablets for CYP that are not catered for in an ‘adult-oriented’ pharmaceutical industry [35]. There has been a global recognition for the need to improve access to patient-appropriate formulations for paediatrics. While there has been a lot of research that covers the technical challenge of producing DFs using 3DP and its quality control [36,37,38], the acceptability of the 3D printed DFs requires investigation. The study also highlights the potential of CYP themselves as co-designers of personalised DFs that could be produced by 3DP to meet their own needs of acceptable and age-appropriate medicines.

The current study provides a proof of concept that 3DP tablets can be administered to CYP. It also provides preliminary data on the acceptability domains of 3DP tablets that can be assessed using established methods. The study generated preliminary data on the acceptability to CYP of directly administered 3DP tablets. Further studies with a larger number of participants will be conducted to assess the impact of tablet size and provide a comparison to established DFs.

## 5. Conclusions

This is the first study to administer 3DP tablets to CYP and report on swallowability for FDM 3D printed tablets in children. The study confirmed several acceptability attributes in CYP. These outcomes were comparable to widely used conventional film-coated biconvex tablets of corresponding sizes. When participants who swallowed both 3D printed and film-coated tablets were asked, on their preference, the most popular response was that both felt ok (43%). This study strongly reveals the clinical potential of FDM-based 3D printed tablets as a viable patient-centric option for CYP. In the future, low-cost FDM 3D printers could be placed as decentralised point-of-care manufacturing units for personalised DFs for CYP. This report provides a small-size feasibility study with a limited number of participants, but importantly takes forward the field of printed medicines by demonstrating direct administration in vivo. To confirm these preliminary findings, studies with a larger number of participants are needed. The results from this feasibility study will inform a larger, definitive study assessing the acceptability of 3DP tablets in children.

## Figures and Tables

**Figure 1 pharmaceutics-14-00516-f001:**
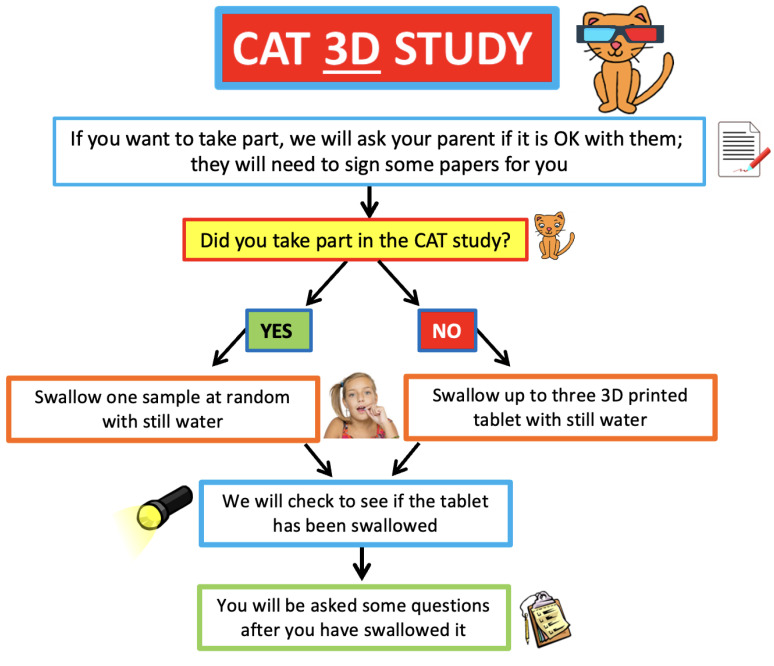
Flow chart of acceptability trials of film-coated and 3D printed tablets.

**Figure 2 pharmaceutics-14-00516-f002:**
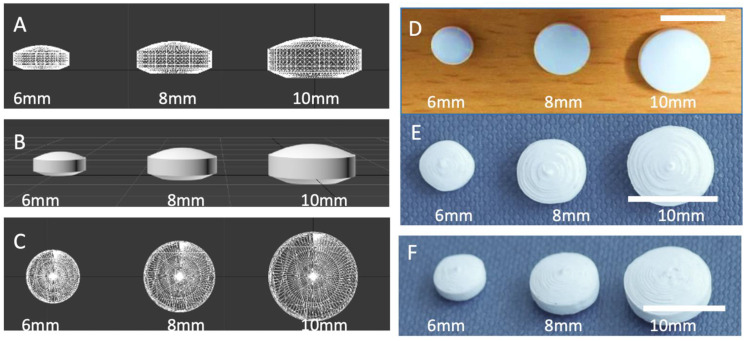
(**A**) Side and (**B**) upper view of a lattice image and (**C**) rendered image of CAD design of convex tablets with three different sizes; (**D**) photograph of film-coated tablets, photographs of (**E**) top and (**F**) view of 3D printed tablets.

**Figure 3 pharmaceutics-14-00516-f003:**
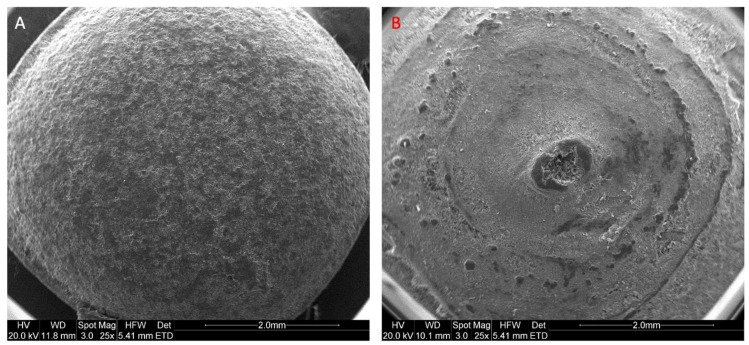
SEM image of a top view of (**A**) film-coated tablets and (**B**) 3D printed tablets.

**Figure 4 pharmaceutics-14-00516-f004:**
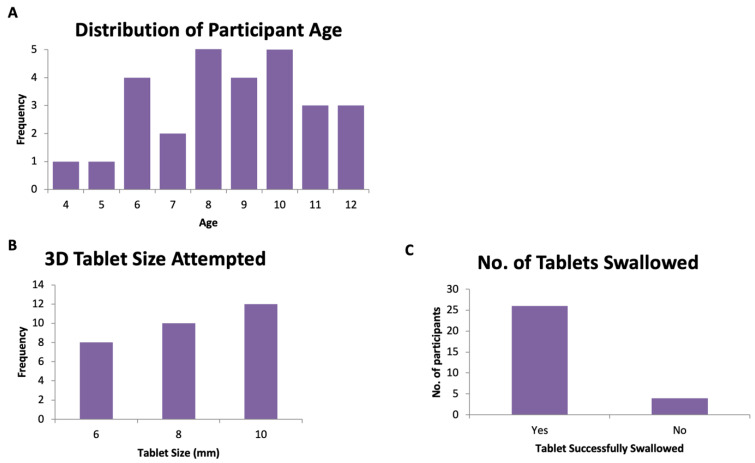
(**A**) Distribution of participants’ age, (**B**) the number of participants versus tablet size, and (**C**) the overall number of tablets swallowed.

**Figure 5 pharmaceutics-14-00516-f005:**
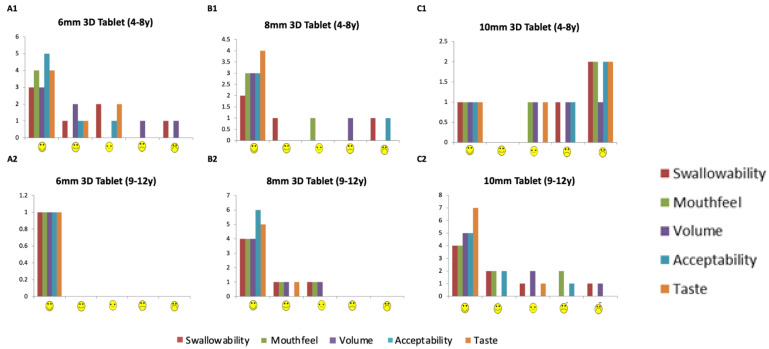
Overall acceptability of a 3D printed tablet. Distribution of reporting using a hedonic scale for the overall acceptability of a 3D printed tablet.

**Figure 6 pharmaceutics-14-00516-f006:**
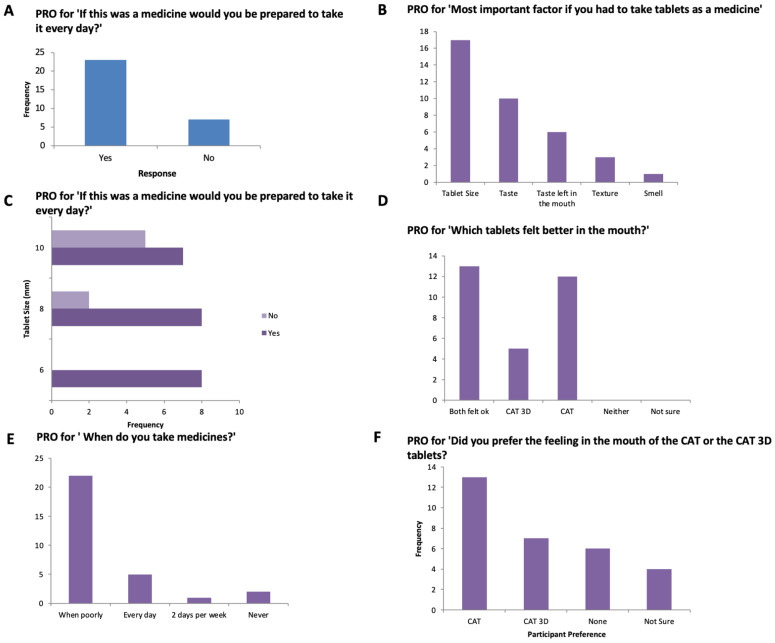
PRO score for (**A**) ‘If this was a medicine would you be prepared to take it every day, (**B**) Most important factor if you had to take tablets, (**C**) ‘If this was a medicine would you be prepared to take it every day, (**D**) Which tablets felt better in the mouth?’, (**E**) When do you take your medicine’, (**F**) ‘Did you prefer the feeling in the mouth of the CAT or the CAT 3D tablets?’.

**Figure 7 pharmaceutics-14-00516-f007:**
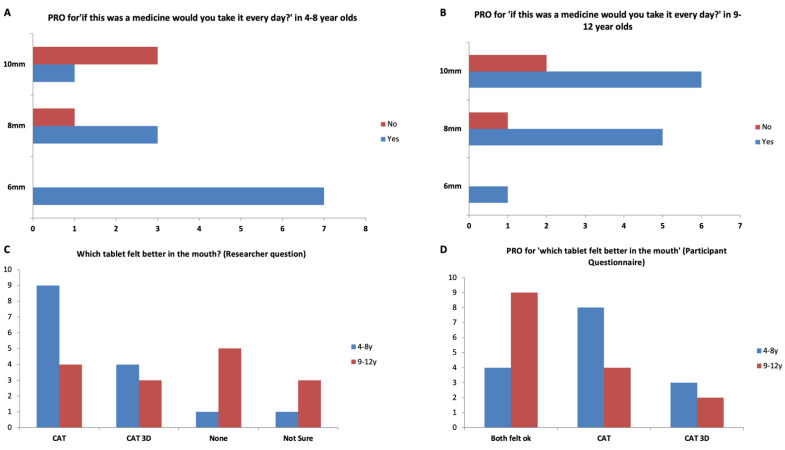
PRO score for ‘if this was a medicine would you take it every day?’ in (**A**) 4–8-year-olds, and (**B**) 9–12 years old, (**C**) Which tablet felt better in the mouth? (Researcher question), and (**D**) ‘Which tablet felt better in the mouth?’ for participants aged 4–8 and 9–12 years.

**Table 1 pharmaceutics-14-00516-t001:** Summary of acceptability of 3D printed tablets *****.

Size of 3D Printed Tablet	6 mm	8 mm	10 mm
**Swallowability**			
Participants aged 4–8 years able to swallow the placebo tablet	7 (100%)	3 (75%)	2 (50%)
Participants aged 9–12 years able to swallow the placebo tablet	1 (100%)	6 (100%)	7 (87.5%)
Rated acceptable on a hedonic scale by participants aged 4–8 years	6 (87.5%)	3 (75%)	1 (25%)
Rated acceptable on a hedonic scale by participants aged 9–12 years	1 (100%)	6 (100%)	7 (87.5%)
**Taste**			
Rated acceptable on a hedonic scale by participants aged 4–8 years	7 (100%)	4 (100%)	2 (50%)
Rated acceptable on a hedonic scale by participants aged 9–12 years	1 (100%)	6 (100%)	8 (100%)
**The volume of water consumed to swallow each placebo tablet**			
Average volume of water (mL) required by participants aged 4–8 years **±SD, (range).**	27.4 ± 14.9 mL (5–48 mL)	15.3 ± 11.8 mL (8–29 mL)	71.5 ± 38.9 mL (44–99 mL)
Average volume of water (mL) required by participants aged 9–12 years	9 mL **†**	16.7 ± 5.8 mL (11–25 mL)	34 ± 16.6 mL (20–59 mL)
Rated acceptable on hedonic scale by participants aged 4–8 years	5 (71.4%)	3 (75%)	2 (50%)
Rated acceptable on hedonic scale by participants aged 9–12 years	1 (100%)	6 (100%)	7 (87.5%)
**Overall Acceptability 4–12 years**			
Rated acceptable on hedonic scale	100%	90%	67%
Willing to take the placebo tablet every day if it was a medicine (participants aged 4–8 years)	8/8 (100%)	8/9 (89%)	7/9 (77.8%)

***** Data from children who successfully swallowed the 3DP tablet (26/30 children in the study). **†** Only one participant aged 9–12 years took 6 mm 3D printed tablet.

## Data Availability

Data are available upon reasonable request. All data relevant to the study are included in the article. The full data set is held by the corresponding author, please email with any requests for extra data.

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
