# Peer review of "Creating Acceptable Tablets 3D (CAT 3D): A Feasibility Study to Evaluate the Acceptability of 3D Printed Tablets in Children and Young People"

_pharmaceutics, 2022, doi:10.3390/pharmaceutics14030516_

Round 1

Reviewer 1 Report

The authors present the results of a small acceptability study of differently sized, FDM-3D printed solid oral dosage forms in children and young people as part of a larger acceptability study that also includes regular, film-coated tablets. A total of 30 participants were included and every participant was given a single 3D printed tablet. 87 % of participants successfully swallowed the tablets and the results of a hedonic-scale questionnaire, specific questions and observations are discussed. Data is also compared to findings obtained with coated tablets. This is very good (but could be discussed even more critically).

This publication is important and, despite the small group size, which the author’s also discuss, of relevance for the field of printed medicines. The manuscript is well written and easy to follow. Prior to publication, I have a few comments that should be addressed. 

In the introduction, dosage forms that a allow an incremental or complete individualization of dosage are discussed. However, the very important dosage form of mini-tablets is only superficially discussed. Multiple acceptability studies have demonstrated that (also multiple) mini-tablets show at least equivalent acceptability to liquids and syrups. This is important to reflect.

How was ensured that the researcher staff members had no influence of the perception of the printed tablets? The protocol is not described sufficiently.

A very low layer height of 80 um was chosen to print the dosage forms. The authors hypothesize that the layer structure might be a reason for the difficulty of swallowing. Are further reductions in layer height feasible to improve this point also in light of economic considerations (printing time)? 80 um already requires extensive printing time.

It would be great if the data for swallowability and taste could be included as detailed as the data on acceptability in Figure 5. It could be included in the supplementary information and would allow a look at the detailed results for these important topics. The description and discussion of the results is fine but I think that interested readers would appreciate the additional information.

Other points to address:

Spelling: spaces, commas, or points missing, especially around reference numbers, e.g. lines 53, 89, 136, 148, …

114: designed tablets are bi-convex round tablets but not caplets.

115: SLA (stl) file format? Introduce SLA.

127: IPC could be opened up, same for RF in line 130. The unit of RF power is W, not w. 

141: open up GC.

Figure 1: Very busy Figure. Pharmaceutics is online only so it is easily possible to improve the flow chart.

Figure 2: Please include images from the side of the dosage forms to enable assessment the convex shape.

Table 1: Do you have any data / information why the taste of 10 mm tablets was only acceptable for 2 of the 4 CYPs while smaller tablets showed 100 % acceptability? Are standard deviations or preferrable confidence intervals for the volume of the consumed water available? There should be quite some spread.

343: Different style of reference.

269: It would be better for the readability to also include the ratio of participants that were able to swallow 10 mm tablets. This is indicated for 6 and 8 mm.

Author Response

Review #1

The authors present the results of a small acceptability study of differently sized, FDM-3D printed solid oral dosage forms in children and young people as part of a larger acceptability study that also includes regular, film-coated tablets. A total of 30 participants were included and every participant was given a single 3D printed tablet. 87 % of participants successfully swallowed the tablets and the results of a hedonic-scale questionnaire, specific questions and observations are discussed. Data is also compared to findings obtained with coated tablets. This is very good (but could be discussed even more critically).

This publication is important, and despite the small group size, which the author’s also discuss, of relevance for the field of printed medicines. The manuscript is well written and easy to follow. Prior to publication, I have a few comments that should be addressed. 

We thank the referee for thinking positively of our manuscript. Please find below a point-to-point response to the reviewer’s comments.

In the introduction, dosage forms that a allow an incremental or complete individualization of dosage are discussed. However, the very important dosage form of mini-tablets is only superficially discussed. Multiple acceptability studies have demonstrated that (also multiple) mini-tablets show at least equivalent acceptability to liquids and syrups. This is important to reflect.

Thank you for raising this important point. We have modified the introduction to cover the importance of mini-tablets for oral drug delivery in the revised version of the manuscript.

How was ensured that the researcher staff members had no influence of the perception of the printed tablets? The protocol is not described sufficiently.

Thank you for raising this important point. We have provided additional explanation (Section 2.6.7.). In brief, the researcher administering the tablets was not involved in the tablet manufacturing process and did not comment on the sample administered to the participants  

A very low layer height of 80 um was chosen to print the dosage forms. The authors hypothesize that the layer structure might be a reason for the difficulty of swallowing. Are further reductions in layer height feasible to improve this point also in light of economic considerations (printing time)? 80 um already requires extensive printing time.

We thank the reviewer for raising this valuable point. We agree that reducing the thickness of the layer would result in a finer resolution. In FDM 3D printing, the most common layer thickness is 200 µm. Several 3D printers have been to optimise to achieve a layer thickness as low as 30 µm. Higher resolution is associated with longer printing time-per-tablets.

It would be great if the data for swallowability and taste could be included as detailed as the data on acceptability in Figure 5. It could be included in the supplementary information and would allow a look at the detailed results for these important topics. The description and discussion of the results is fine but I think that interested readers would appreciate the additional information.

Thank you for highlighting this, a legend has now been added on Figure 5 to show the different data points which includes swallowability and taste. This will be clearer for the readers.

Other points to address:

Spelling: spaces, commas, or points missing, especially around reference numbers, e.g. lines 53, 89, 136, 148, …

We apologies for these errors. We have carefully checked the manuscript for grammar errors and typos.
114: designed tablets are bi-convex round tablets but not caplets.

Many thanks for spotting this. We have changed the shape of the tablet to bi-convex.

115: SLA (stl) file format? Introduce SLA.

We have introduced the acronym in the revised version of the manuscript

127: IPC could be opened up, same for RF in line 130. The unit of RF power is W, not w. 

Many thanks for spotting this. We have corrected these as advised in the revised version of the manuscript.

141: Open up GC.

Many thanks for spotting this. We have corrected this as advised in the revised version of the manuscript.

Figure 1: Very busy Figure. Pharmaceutics is online only so it is easily possible to improve the flow chart.

Thank you, we have simplified Figure 1 in the revised version of the manuscript.

Figure 2: Please include images from the side of the dosage forms to enable assessment the convex shape.

Thank you these have now been included

Table 1: Do you have any data / information why the taste of 10 mm tablets was only acceptable for 2 of the 4 CYPs while smaller tablets showed 100 % acceptability? 

Thank you for raising this important point unfortunately we do not have any further data as to why this is, one hypothesis is that perhaps because the younger children had some difficulty swallowing the tablet, they retained the table tin their mouth for longer which may have led to them being more aware of the taste, but we have no further patient comments unfortunately to include or validate this.

Are standard deviations or preferrable confidence intervals for the volume of the consumed water available? There should be quite some spread.

We added the standard deviation and range for each value. We have identified an error during transposing the data, this has now been rectified in the revised version of the manuscript.

343: Different style of reference.

Apologies, this has been amended in the revised version of the manuscript.

269: It would be better for the readability to also include the ratio of participants that were able to swallow 10 mm tablets. This is indicated for 6 and 8 mm.

This section has been modified as advised by the referee.

Reviewer 2 Report

The author indicates he/she has performed heavy metal assay by ICP. There is no data to support this claim.

On page 10, the label for the 10 mm tablet is missing 3D on the first graph and 10 mm on the second graph

Author Response

The author indicates he/she has performed heavy metal assay by ICP. There is no data to support this claim.

A heavy metal assay of 3D printed tablets is available in Table S1 in Supplementary Data document.

On page 10, the label for the 10 mm tablet is missing 3D on the first graph and 10 mm on the second graph

Apologies, this has been amended in the revised version of the manuscript.

Reviewer 3 Report

thank you for the opportunity to review such an interesting paper, I think it is of interest to the reader as it opens new research venues. 

I think conclusion should be more focused on the implications of the findings, instead of simply repeating them. 

Some sentences begin with number or percetanges instead of words,  think it should be checked.

Author Response

Thank you for the opportunity to review such an interesting paper, I think it is of interest to the reader as it opens new research venues. 

We thank the reviewer for very positive feedback.

I think conclusion should be more focused on the implications of the findings, instead of simply repeating them. 

We have revised the conclusion to reflect the point raised by the referee.

Some sentences begin with number or percentages instead of words, think it should be checked.

Thank you for highlighting this, apologies this has been corrected in the revised version.